# HPV Vaccination in Women Treated for Cervical Intraepithelial Neoplasia: A Budget Impact Analysis

**DOI:** 10.3390/vaccines9080816

**Published:** 2021-07-22

**Authors:** Michele Basile, Giovanna Elisa Calabrò, Alessandro Ghelardi, Roberto Ricciardi, Rosa De Vincenzo, Americo Cicchetti

**Affiliations:** 1Graduate School of Health Economics and Management (ALTEMS), Università Cattolica del Sacro Cuore, 00168 Rome, Italy; michele.basile@unicatt.it (M.B.); americo.cicchetti@unicatt.it (A.C.); 2Section of Hygiene, University Department of Life Sciences and Public Health, Università Cattolica del Sacro Cuore, 00168 Rome, Italy; 3VIHTALI (Value in Health Technology and Academy for Leadership & Innovation), Spin-Off of Università Cattolica del Sacro Cuore, 00168 Rome, Italy; robertoricciardi.mail@gmail.com; 4Azienda USL Toscana Nord-Ovest, UOC Ostetricia e Ginecologia, Ospedale Apuane, Via Enrico Mattei, 21, 54100 Massa, Italy; ghelardi.alessandro@gmail.com; 5Gynecologic Oncology Unit, Fondazione Policlinico Universitario A. Gemelli, IRCCS, Dipartimento Scienze della Salute della Donna, del Bambino e di Sanità Pubblica, 00168 Rome, Italy; rosa.devincenzo@unicatt.it; 6Dipartimento di Scienze della Vita e Sanità Pubblica, Università Cattolica del Sacro Cuore, 00168 Rome, Italy

**Keywords:** adjuvant HPV vaccination, cervical intraepithelial neoplasia, budget impact analysis, health technology assessment

## Abstract

Human Papillomavirus (HPV) is the most common sexually transmitted infection. Its progression is related to the development of malignant lesions, particularly cervical intraepithelial neoplasias (CINs). CINs correlate with a higher risk of premature births, and their excisional and ablative treatment further increases this risk in pregnant women. These complications are also correlated with higher healthcare costs for their management. In Italy, more than 26,000 new cases of CINs are estimated to occur yearly and their economic burden is significant. Therefore, the management of these conditions is a public health priority. Since HPV vaccination is associated with a reduced risk of relapse in women surgically treated for HPV-related injuries, we estimated the economic impact of extending HPV vaccination to this target population. This strategy would result in a significant reduction in the general costs of managing these women, resulting in an overall saving for the Italian Health Service of €155,596.38 in 5 years. This lower cost is due not only to the reduced incidence of CINs following vaccination, but also to the lower occurrence of preterm births. Extending HPV vaccination to this target population as part of a care path to be offered to women treated for HPV injuries is therefore desirable.

## 1. Introduction

Human papilloma virus (HPV) infection is very frequent in the population and it is mainly transmitted sexually. Based on sequence information collected by the International HPV Reference Center more than 225 HPV types have been described [1]. According to their ability to cause pre-cancerous lesions, these are categorized as high-risk (HR) and low-risk (LR) HPV types. Most infections diagnosed are due to HR-HPV genotypes, namely, HPV 16, 18, 31, 33, 35, 39, 45, 51, 52, 56, 58, 59, 68, 73, and 82. Specifically, the HPV genotypes 16 and 18 are related to approximately 70% of all cervical cancers (CC) worldwide, and types 31, 33, 45, 52, and 58 cause a further 20% [2]. HPV infection is more common in adolescent and young adult girls, peaking at about 20% in women aged 20 to 24 years and subsequently declining in women older than 30 years [2]. In most cases (about 90%) the infection resolves spontaneously, with viral clearance, within two years; by contrast, the persistence of HR-HPV infection can lead to dysplasia and an increased risk of developing cancer [3]. One of the most frequent precancerous precursor lesions associated with the persistence of HPV infection is cervical intraepithelial neoplasia (CIN). In terms of the natural history of CIN, CIN1 regresses in about 60% of cases, persists in 30%, progresses to CIN 3 in 10%, and to invasion in 1%. The rate of progression of CIN 2+ to invasive CC is higher; for this reason, the management of CIN 2+ is different from that of CIN1 [4]. CINs correlate with a higher risk of premature births, and excisional and ablative treatment of these lesions further increases this risk in pregnant women. The frequency and severity of obstetric complications (spontaneous preterm birth, low birth weight, admission to neonatal intensive care, perinatal mortality, premature rupture of the membranes, and chorioamnionitis) increase in relation to the increased depth of the cone. These complications are also more associated with excision than ablation and are correlated with higher healthcare costs for their management [5,6,7,8]. In Italy, 21,308 cases of CIN1, 3218 of CIN2, and 3,518 of CIN3 are estimated to occur annually [5], and in 2020, 2,400 new cases of CC were estimated [6]. CC is the fourth most common cancer among women worldwide, with an estimated 570,000 new cases and 311,365 deaths in 2018. Moreover, worldwide the burden of CC is estimated to increase to 700,000 cases and 400,000 deaths in 2030, with similar rises projected in subsequent years [9]. According to the World Health Organization (WHO) data, almost all CC cases (99%) are linked to HR-HPV infection and the effective primary (HPV vaccination) and secondary prevention approaches (screening for, and treating precancerous lesions) can prevent most CC cases [10]. Since 2017, the WHO has recommended that HPV vaccination be comprised in national immunization programs. Since greater benefit and protection are thought to be achieved by administering the vaccine to subjects naïve to HPV infection, the WHOs primary target of vaccination is females aged 9–14 years, ideally before their sexual debut. Secondary targets, such as females >15 years old, males, and high-risk individuals (e.g., HIV-positive subjects) could also be considered if the strategy is affordable and sustainable [11]. By 2020, most European countries had introduced HPV vaccination into their national immunization programs [12]. In Italy, HPV vaccination has been recommended and actively offered free of charge to girls in their 12th year of life (11 years old) in all Italian regions since 2007–2008. In addition, some regions have extended the active offer and/or the free-of-charge condition to girls of other age groups. In accordance with the 2017–2019 National Vaccine Prevention Plan (PNPV 2017–2019) and the new “Essential Levels of Care” (LEA), free vaccination during the 12th year of age has also been provided for males, starting from the 2006 cohort, for men who have sex with men (MSMs) and for immunocompromised patients (e.g., HIV) [13].

Three vaccines are currently available: a bivalent vaccine (Cervarix^®^) that targets HR-HPV 16 and 18; a quadrivalent vaccine (Gardasil^®^) that includes the LR-HPV types 6 and 11, in addition to the HR-HPV 16 and 18; and the latest nine-valent vaccine (Gardasil 9), which targets types 31, 33, 45, 52, and 58 as well as the same HPV types as the quadrivalent vaccine. These vaccines are licensed to prevent cervical, vulvar, vaginal, and anal cancers. Additionally, the quadrivalent and nine-valent vaccines prevent genital warts [2]. As the nine-valent vaccine targets five more HPV types than its quadrivalent predecessor, it can potentially prevent 70–90% of cervical, vulvar, vaginal, and anal cancers, and 45–80% of precancerous injuries of the cervix [14].

Two recent meta-analyses [15,16] have suggested that HPV vaccination in women undergoing surgery for HPV-related disease could impact on disease recurrence. Furthermore, the National Guidelines System of the Italian National Health Institute has recently published the Guidelines for the prevention of CC which contains a specific recommendation on HPV vaccination in women treated for CIN 2/3 [17]. It is therefore important to discuss the potential benefits of vaccination in this specific set of patients. 

In order to ensure adequate protection for all citizens, it is necessary to apply the vaccination appropriateness principle and evaluate the healthcare system sustainability. In this regard, Health Technology Assessment (HTA) represents a rigorous and evidence-based tool to support the processing and implementation of appropriate, effective, and sustainable immunization strategies [18]. In Italy, the 2012–2014 National Immunization Program (NIP) defined criteria for the evaluation of new and currently used vaccines to be included in the NIP [19]. These criteria, still applied, are based on the multidisciplinary process of HTA, “that summarizes evidence about medical, social, ethical, and economic issues related to health technology in a systematic, clear, impartial, and strong way. Its purpose is to support the definition of effective, safe, patient-centered health policies and oriented to achieve best value” [20]. Economic evaluation supports HTA in order to inform policymakers of the value to society and to the healthcare system conferred by a given allocation of resources. In accordance with the HTA methodological framework for the production and systematization of evidence, defined as the “HTA core model”, of the European network for Health Technology Assessment (EUnetHTA), a fundamental feature of the assessment process is the economic aspect [21]. 

The economic burden of HPV-related diseases is significant. HPV vaccination is associated with a reduced risk of relapse in women surgically treated for HPV-related injuries. Therefore, our analysis aims to estimate the financial impact due to the extension of HPV vaccination with a nine-valent vaccine (9vHPV) to women previously treated for HPV-related lesions. From a public health perspective, we also included in the analysis the costs related to obstetric complications potentially avoidable with vaccination in this target population in which the risk of preterm birth is greater.

## 2. Materials and Methods

A Budget Impact Analysis (BIA) was made from the National Health Service (NHS) perspective, in order to estimate the financial impact due to extension of HPV vaccination with 9vHPV to women surgically treated for HPV-related injuries. The BIA included the following input data:Population eligible for HPV vaccination (target population);Epidemiology of HPV-related precancerous lesion in Italy;Vaccine costs and estimated vaccination coverage;Costs of management and follow-up of HPV-related lesions;Number of estimated pregnancies in women with HPV-related lesions, by age group;Incidence rate of obstetric complications in the target population and costs of the preterm pregnancy management.

The analysis considered a 5-year time horizon. The input data of the economic model were validated by two clinicians with proven experience in the management of women with HPV-related lesions. 

In order to assess the robustness of the results obtained in the analysis, a one-way sensitivity (OWA) analysis was developed to determine the drivers whose variation affects the most estimations achieved in the base-case scenario. Each parameter included in the OWA was associated with a level of uncertainty equal to 25% of its average value.

The results are shown in terms of the difference in resource consumption between two alternative scenarios (with and without vaccination).

### 2.1. Target Population, Epidemiology of HPV-Related Precancerous Lesion in ITALY and Vaccine Costs

The target population was identified on the basis of previous epidemiological studies [22,23], the incidence data from which were adapted to the Italian setting. The target population was calculated on the basis of the Italian population in 2020 and consisted of 12,097 women suffering from CINs (in particular, persistent CIN1, CIN2, and CIN3) (Table 1).

The assumptions applied in the economic model regarding the cost of the vaccine, vaccine strategy (*n*° of doses), and vaccination coverage are shown in Table 2. The probability of relapse in both vaccinated and unvaccinated women treated for HPV-related lesions was defined according to the SPERANZA study [25].

The results of the analysis are expressed in terms of the difference in resource consumption between two alternative scenarios:Scenario 1: this does not include post-treatment vaccination and considers women with a 6.4% probability of relapse [25];Scenario 2: this considers post-treatment vaccination and women who, owing to the effectiveness of the vaccine, have a 1.2% probability of relapse [25].

Table 3 shows the initial populations of the two scenarios under analysis.

Within the decision-making model in both scenarios, the population flows in a probabilistic tree built on the natural history of the illness over the period considered (5 years).

### 2.2. Costs of the Management and Follow-Up of HPV-Related Lesions 

Regarding the management of HPV-related lesions, different frequencies and percentages of use were applied to treatments, according to the severity of the lesions (Table 4). Specifically, in the management of CIN1 lesions, as reported in the study by Rossi et al. [23], we considered the following interventions and percentages of use: Loop Electrosurgical Excision Procedure (LEEP) or surgical conization (45.14%), laser vaporization (38.95%), and diathermocoagulation (15.91%). Thus, the average weighted cost of CIN1 management is €693.51. For the management of CIN 2–3, the analysis considered LEEP treatment based on expert opinion. To determine the average cost associated with the management of each type of lesion, the Italian Tariff of Outpatient Specialist Services (Tariffario delle prestazioni specialistiche ambulatoriali) [26] was used. The provision of care for the lesions management was associated with the cost of the Diagnosis Related Group (DRG) 364, considering the Day Hospital (DH) regime and equal to €1,019.00.

On the basis of the expert opinion regarding women undergoing treatment, an annual frequency of two follow-up visits (one every six months) was assumed [25]. The cost of the scheduled examinations (pap test, HPV test, and colposcopy) was thus calculated regardless of the lesion severity, the estimated average annual cost being €207.00 (Table 5). Furthermore, it was assumed that, starting from the third year following treatment, the number of pap tests, HPV tests, and colposcopies would be reduced to one per year [25].

### 2.3. Estimated Pregnancies in Women with HPV-Related Lesions, Incidence Rate of Obstetric Complications in the Target Population and Costs of the Preterm Pregnancy Management

CINs correlate with a higher risk of premature births, and excisional and ablative treatment of these lesions further increases this risk in pregnant women. The obstetric complications are also more associated with excision than ablation and are correlated with higher healthcare costs for their management [5,7,8]. In order to determine the overall impact also in terms of the number of preterm pregnancies in the target population, the model took into consideration the population groups of childbearing age and the fertility rate of each of these groups (Table 6). This calculation yielded 324 cases of pregnancy.

Regarding the costs related to disease management and the follow-up after discharge of the premature pediatric patient, reference was made to the study by Cavallo et al. [29] which takes into consideration the direct costs incurred by the NHS, the out-of-pocket costs incurred by the patient, and productivity losses in the following periods:From birth to discharge;The first 6 months after discharge;From 6 months post-discharge and onwards.

These costs, referring to the year 2015, were discounted by applying the Italian Institute of Statistics (Istituto Italiano di Statistica–ISTAT) index of 1.01 (Table 7). A cost of €32,784.60 was attributed to the management of each preterm birth [29]. These costs were associated with the onset rates of the complications considered, in accordance with those referring to the general population extrapolated from the 2015 Società Italiana di Ginecologia e Ostetricia (SIGO) Guidelines [30] increased by 20% [31] to reflect the increased risk of preterm pregnancy in women with HPV-related lesions (Table 8).

The estimated number of pregnancies in women with HPV-related lesions is 324. The number of preterm births is 26 and 6 in the “without” and “with vaccination” scenarios, respectively. Over these number of cases the following costs have been weighed: the initial hospitalization costs (€20,707.02), the traveling costs (€2375.52), the productivities losses (€9702.06), and the general societal costs (€25,893.37).

## 3. Results

On the basis of the costs calculated according to the methodology described and the distribution of women among the arms of the probabilistic trees in the two scenarios analyzed, the results yielded by the model demonstrate that extending HPV vaccination with 9vHPV to women with a previous HPV-related lesion would yield a significant reduction in the general costs of managing patients from the second year of analysis onwards, the saving in the second year being −€389,803.89 over the whole sample of patients considered

The overall savings in the period of the analysis (5-year time horizon) are €155,596.38 (Table 9). This lower cost for the Italian Health Service is due not only to the reduced incidence of CINs following vaccination, but also to the lower occurrence of preterm births. In particular, the absorption of NHS resources in the “without vaccination” scenario in 5 years is €3,015,431.51 (relapse costs = €1,510,476.33 + costs of preterm births management = €1504. 955.18). Instead, the absorption of NHS resources in the “with vaccination” scenario in 5 years is €285,983.13 (relapse costs = €344,577.41 + costs of preterm births management = €343,317.90 + vaccine costs = €2,171,939.82).

Specifically, comparison between the resources absorbed in the two scenarios revealed the greatest difference in the second year of the analysis, when the costs fell from €505,009.09 in the scenario without vaccination to €115,205.20 in the vaccination scenario; also, this difference is mainly due to the lower numbers of both relapses and premature births in the vaccination scenario.

The study also implemented a univariate sensitivity analysis (OWA) in order to determine the robustness of the results obtained and to characterize the uncertainty surrounding the parameters. Specifically, a standard deviation of 25% was assumed and assigned to each parameter in the BIA.

This analysis showed that the parameter that had the greatest impact on the variance of the results was the number of the vaccine doses, followed by the relapse rate of HPV-related lesions (Figure 1).

As can be seen from Figure 1, the other parameters considered in the deterministic sensitivity analysis displayed a lower level of uncertainty, and their variation had a marginal impact on the results of the analysis.

## 4. Discussion

In our study, we implemented a BIA to estimate the impact, in terms of the absorption of healthcare resources, of extending HPV vaccination with the 9vHPV to women undergoing surgery for CINs. These cervical HPV-related lesions correlate with a higher risk of preterm births, and excisional and ablative treatment further increases this risk in pregnant women. The obstetric complications of the CINs are also more associated with excision than ablation and are correlated with higher healthcare costs for their management. From a public health perspective, HPV vaccination constitutes a fundamental strategy for the primary prevention of CC and other HPV-related lesions. As such, it justifies the resources allocated for the purpose of achieving a given degree of coverage among specific population cohorts, including women previously treated for HPV-related lesions and at risk of developing new infections and post-treatment relapses. From a public health perspective, we also included in our analysis the costs related to obstetric complications potentially avoidable with vaccination in women already treated for CINs in which the risk of preterm birth is greater.

Our model considered the perspective of the Italian NHS and a 5-year time horizon. The input of the economic model was validated by two gynecologists with proven experience in the management of this target population. 

Two recent meta-analyses [15,16] suggested that HPV vaccination in women undergoing surgery for HPV-related disease could impact on disease recurrence. Therefore, it is realistic to discuss the potential benefits of vaccinating this specific set of patients.

The aim of these meta-analyses was to analyze the available evidence supporting the use of prophylactic HPV vaccines to reduce the recurrence risk of CIN after surgical treatment. The meta-analysis by Bartels et al. [15] included five studies with 3562 patients selected for analysis (1453 patients in the vaccinated group and 2109 in the placebo or unvaccinated group). The incidence of histologically-confirmed CIN2+ was reduced in the vaccinated compared with the unvaccinated group (OR 0.51, 95% CI 0.35 to 0.74, *p* = 0.0003). The number needed to treat (NNT) in order to prevent one recurrence was 43. Both pre-treatment vaccination (OR 0.48, 95% CI 0.25 to 0.94, *p* = 0.03, NNT 38) and adjuvant vaccination (OR 0.53, 95% CI 0.34 to 0.81, *p* = 0.004, NNT 40) reduced recurrence rates. These results confirmed that adjuvant HPV vaccination reduces the risk of recurrent CIN2+.

A meta-analysis by Jentschke et al. [16] included 10 studies, five of which were those considered by Bartels et al., and analyzed the effect of pre- or post-conization vaccination against HPV. The overall study population included 21,059 patients (3909 vaccinated vs. 17,150 controls). The results showed an important reduction in the risk of developing new high-grade intraepithelial lesions after HPV vaccination (relative risk (RR) 0.41; 95% CI [0.27; 0.64]), regardless of HPV type. Owing to the heterogeneity of the study populations, various sub-analyses concerning HPV type, patient age, vaccination time, and follow-up have been carried out. The age-dependent analysis presented no differences between women under 25 years of age (RR 0.47; 95% -CI [0.28; 0.80]) and older women (RR 0.52; 95% -CI [0.41; 0.65]). Regarding HPV 16/18-positive CIN2+, the results presented a RR of 0.37 (95% CI [0.17; 0.80]). Overall, the number of women to be vaccinated before or after surgery in order to prevent one case of recurrent CIN 2+ (NNT) was 45.5. Therefore, this meta-analysis also showed a substantial reduction in the risk of developing recurrent CIN after conization and HPV vaccination in comparison with conization only.

From a societal perspective, HPV vaccination constitutes a key primary prevention opportunity with regard to CC and many other HPV-related lesions. Indeed, the persistence of HPV can lead to the onset of HPV-related precancerous lesions. There are currently several treatment strategies for patients who develop an HPV-related lesion, including LEEP, laser vaporization, diathermocoagulation, cryotherapy, and surgical conization, which are used with different frequencies depending on the stage of the lesion to be treated [23]. Furthermore, HPV-related lesions also correlate with a greater risk of preterm births [5]. Kyrgiou et al. [7,8] report that women with CINs have a higher baseline risk for prematurity, and excisional and ablative treatment further increases that risk. The frequency and severity of adverse sequelae increases with increasing cone depth and is higher for excision than for ablation. Furthermore, spontaneous preterm birth, premature rupture of the membranes, chorioamnionitis, low birth weight, admission to neonatal intensive care, and perinatal mortality are also significantly increased after treatment for CINs [7,8].

These complications may necessitate complex management of the premature newborn, which increases the burden on NHS resources, raises expenses for families and leads to a considerable loss of productivity for society [31].

In the current context, in which healthcare systems have limited resources available, it is imperative to determine and implement solutions that enable the costs of managing health needs to be reduced without lowering the level of patient care. 

Economic evaluations of vaccines and vaccination usually consider only part of the related benefits; in recent years, however, the scientific community has stressed the importance of considering a wider range of benefits, in order to take into account the vaccination value in its entirety [32]. For instance, in 2005, Bloom et al. stated that exact assessments of the vaccination value must contemplate wider outcomes, including the vaccination effects on children’s cognitive development and their educational achievement, work productivity, income, savings, and also fertility [32]. Early analyses conducted to estimate the greatness of these wider benefits showed that vaccination was considerably underestimated, which has significant consequences for public health and immunization policies worldwide [33]. Therefore, more studies and improved data will be essential in order to implement research into the vaccination value.

To underline the broader value of HPV vaccination, we considered a new target population, i.e., women already treated for HPV-related lesions. Furthermore, in developing our economic model, we included for the first time the costs related to the management of preterm births which are a complication significantly increased after treatment for CINs. Our analysis determined the possible savings obtainable in the case of extending HPV vaccination with a nine-valent vaccine to women treated for previous HPV-related lesions. This extension would lower costs for the Italian NHS by €155,596.38 in the time interval considered (5 years), owing to the reduction in CINs following vaccination and the lower occurrence of preterm births due to HPV-related lesions. This is a conservative result, as it is based on the post-vaccination recurrence rate reported in the study by Ghelardi et al. [25], which considered the quadrivalent anti-HPV vaccine; indeed, the scientific literature has not yet provided efficacy data on the 9vHPV vaccine under analysis. In addition, the SPERANZA study [25] showed that the relapses experienced by women treated for CINs who subsequently underwent HPV vaccination were caused by HPV 82 and 33 serotypes. As the nine-valent vaccine also immunizes against infections due to serotype 33, it is reasonable to assume that the recurrence rate following administration of this vaccine is lower than the 1.2% assumed in the present analysis.

As also explained in the 2017–2019 PNPV [13], the State has the duty to ensure the protection of the health of all citizens, both by treating sick individuals and by safeguarding healthy individuals. According to article 32 of the Italian Constitution, “the Republic defends health as an essential right of citizens and public interest, and assurances free healthcare to the needy”. Vaccination is one of the main prevention tools for Public Health and it is effective, cost/effective, and safe. Institutions should therefore plan adequate vaccination strategies in such a way as to guarantee equal and universal access to vaccination through, for instance, the activation of specific programs for the most vulnerable groups of the population, in compliance with the fundamental rights of citizens. 

In a period in which the sustainability of healthcare systems is one of the most relevant topical issues and in which we are witnessing a continuous evolution of the epidemiological context, demographic change, and major social transformations, prevention is a fundamental public health tool, despite the current dearth of economic resources [34]. However, the evidence of the social value of vaccinations is central to the concept of prevention as a system of investment in health [35].

From a public health perspective, HPV vaccination constitutes a fundamental strategy for the primary prevention of CC and other HPV-related lesions. As such, it justifies the resources allocated for the purpose of achieving a given degree of coverage among specific population cohorts, including women previously treated for HPV-related lesions and at risk of developing new infections and post-treatment relapses [36].

Our economic analysis showed that extending HPV vaccination to women previously treated for an HPV-related precancerous condition of the cervix would result in a lower cost for the NHS, owing not only to the reduced incidence of these lesions following vaccination but also to the lower occurrence of preterm births. Indeed, to our knowledge, the present model is the first to consider, from a public health perspective, also the possible benefits of vaccination in reducing obstetric complications associated with the presence of HPV-related lesions and their treatment. Indeed, such lesions result in a greater risk of preterm births, more complex management of premature newborns, and a heavier burden on the NHS resources.

## 5. Conclusions

HPV is the most common sexually transmitted infection and its progression is related to the development of malignant lesions especially of the uterine cervix. In fact, according to the WHO data, almost all CC cases are linked to HR-HPV infection. For this reason, WHO launched a “call to action” for the elimination of CC. In order to attain this goal, the WHO outlined the necessary actions in its global strategy: prevention through vaccination; screening and treatment of precancerous lesions; treatment and palliative care for invasive cervical cancer. These three pillars must be implemented collectively and at scale to achieve the goal of elimination [9].

Therefore, the HPV-related diseases are one of the global public health priorities and their economic burden is also relevant. 

HPV vaccination is associated with a reduced risk of relapse in women surgically treated for HPV-related injuries and it is important to evaluate the potential benefits of vaccination also in this specific set of patients. Our analysis estimated the financial impact due to the extension of HPV vaccination with the 9vHPV to women previously treated for CINs. From a public health perspective, we also included in the analysis the costs related to obstetric complications potentially avoidable with vaccination in this target population in which the risk of preterm birth is greater. This strategy would result in a significant reduction in the general costs of managing these women, resulting in an overall saving for the Italian Health Service of €155,596.38 in 5 years. This lower cost is due not only to the reduced incidence of CINs following vaccination, but also to the lower occurrence of preterm births. 

In light of the results obtained, we can affirm that vaccination with the nine-valent HPV vaccine should be potentiated in the context of prevention in Italy and also that it should be extended to women treated for HPV-related lesions.

This strategy could be an action to be implemented for the elimination of CC through vaccination around the world.

## Figures and Tables

**Figure 1 vaccines-09-00816-f001:**
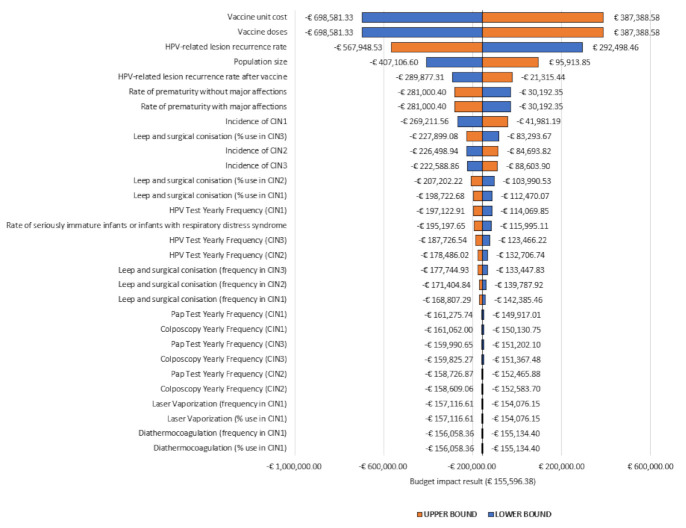
Deterministic analysis.

**Table 1 vaccines-09-00816-t001:** HPV-related precancerous lesion population.

Italian Female Population Aged 25–64 Years (1 January 2020) = 16,232.654 [22]			
CIN	Incidence of CIN	Cases of CIN	Source
CIN 1	0.089%	4354 *	[23]
CIN 2	0.023%	3701	[23]
CIN 3	0.025%	4042	[23]
Total Italian women with precancerous lesions (CINs) per year		12,097	

* The eligible population consisted only of patients with persistent CIN-1 lesions (equal to 30%) [24].

**Table 2 vaccines-09-00816-t002:** Parameters considered in the economic model and related to vaccine cost, vaccination strategy (*n*° of doses), and estimated vaccination coverage.

Vaccine Cost	€63/dose
Number of doses	3
Estimated vaccination coverage	95%

**Table 3 vaccines-09-00816-t003:** Initial populations of the two scenarios under analysis.

Total Women with Precancerous Lesions/Year = 12,097			
Scenario	Parameters for Calculating Target Population in Each Scenario	Calculation by Target Population	*N*° Women (per Year) per Scenario
Scenario 1 “without vaccination”	Women with precancerous lesions per year × Probability of post-treatment HPV-related lesion recurrence ^1^	12,097 × 6.4%	774
Scenario 2 “with vaccination”	Women with precancerous lesions per year × Probability of post-treatment HPV-related lesion recurrence ^1^ × Percentage of women not covered by vaccine ^2^	(12,097 × 6.4%) × 5% = 39 women	177
	Women with precancerous lesions per year × Probability of HPV-related lesion recurrence after HPV vaccination, post-treatment ^3^ × % vaccination coverage equal to 95%	(12,097 × 1.2%) × 95% = 138 women	

^1^ Equal to 6.4% (source: SPERANZA study [25]); ^2^ Equal to 5% (in the model, a vaccination coverage of 95%); ^3^ Equal to 1.2%. (Source: SPERANZA study [25]).

**Table 4 vaccines-09-00816-t004:** Unit cost of therapeutic alternatives for the management of HPV-related lesions and percentage of use per treatment by degree of lesion.

		CIN 1		CIN 2		CIN 3	
Treatment	Unit Cost	% Use [22]	Frequency	% Use *	Frequency	% Use *	Frequency
Leep and surgical conization	€1469.00 [27,28]	45.14%	1	100.00%	1	100.00%	1
Laser vaporization	€60.00 [27]	38.95%	1	0.00%	0	0.00%	0
Diathermocoagulation	€44.64 [26]	15.91%	1	0.00%	0	0.00%	0
**Total**		**100.00%**	**€693.51**	**100.00%**	**€1469.00**	**100.00%**	**€1469.00**

* Expert Opinion.

**Table 5 vaccines-09-00816-t005:** Annual cost of follow-up by grade of HPV-related lesion.

Test	Unit Cost of Test	N. of Test for CIN 1 per Year	N. of Test for CIN 2 per Year	N. of Test for CIN 3 per Year
Pap test	€11.16 [28]	2	2	2
HPV Test	€81.60 [23]	2	2	2
Colposcopy	€10.74 [28]	2	2	2
Annual cost per CIN type		€207.00	€207.00	€207.00

**Table 6 vaccines-09-00816-t006:** The number of estimated pregnancies in women with HPV-related lesions, by age group.

**Born in 2020** [22]				420,084
Italian Population 2020 (Females) (25–49 Years) [22]				9,417,183
Rate in Females Aged 25–49 Years				4.46%
Age Range	N°. of Women by Age-Group [22]	Women of Childbearing Age with HPV-Related Lesions	Weighted Birth Rate *	N° of Estimated Pregnancies in Women with HPV-Related Lesions
25–29	1,523,050	1135	7.88%	89
30–34	1,623,692	1210	10.35%	125
35–39	1,779,210	1326	6.40%	85
40–44	2,098,266	1564	1.43%	22
45–49	2,392,965	1783	0.11%	2
	Total	7390		324

* Birth rate weighted by fertility rate and age group (standardized to 100%).

**Table 7 vaccines-09-00816-t007:** Preterm birth management costs (pre- and post-discharge) [29].

Preterm Birth Management-Pre-Discharge		
Costs Incurred by the National Health Service		
Initial		€20,707.02
Costs Incurred by the Patient		
Travel		€2375.52
Productivity losses		€9702.06
Total cost of initial hospitalization (from birth to discharge)		€32,784.60
Management of Preterm Birth-Post-Discharge		
Total Cost Incurred by the National Health Service	Costs (1–6 months)	Costs (6 months ->)
Other hospitalizations	€811.03	€470.66
Medical consultations (pediatrician)	€71.71	€78.78
Specialist visits (follow-up on an outpatient basis)	€90.90	€74.74
Other specialist visits	€53.53	€33.33
Medicines	€283.81	€105.04
Lab tests	€14.14	€10.10
Imaging tests	€107.06	€33.33
Rehabilitation therapy	€10.10	€13.13
Psychological support	€5.05	€9.09
Total costs	€1447.33	€828.20
Costs Incurred by Families	Costs (1–6 months)	Costs (6 months ->)
Medicines	€271.69	€110.09
Pediatric visits	€57.57	€132.31
Other specialist visits	€5.05	€4.04
Laboratory and imaging tests	€1.01	€1.01
Rehabilitation therapy	€-	€-
Informal assistance (with shared fees)	€164.63	€1194.83
Informal assistance (without shared fees)	€1121.10	€3653.17
Travel expenses	€96.96	€69.69
Productivity losses	€10,348.46	€6386.23
Total costs incurred by households and productivity losses	€12,066.47	€11,551.37
Total social cost	€13,513.80	€12,379.57
	€25,893.37	

**Table 8 vaccines-09-00816-t008:** Incidence rate of obstetric complications over the number of estimated pregnancies in women with CINs (*n* = 324).

	Incidence Rate [Source]	N. of Preterm Births in “without Vaccination” Scenario	N. of Preterm Births in “with Vaccination” Scenario
Infants who are severely immature or with respiratory distress syndrome	1.08% [30,31]	3	1
Prematurity with major complications	3.42% [30,31]	11	3
Prematurity without major complications	3.42% [30,31]	11	3
Total	7.92%	26	6

**Table 9 vaccines-09-00816-t009:** Absorption of NHS resources in the “without vaccination” scenario, “with vaccination” scenario, and differential analysis.

**“Without Vaccination” Scenario**						
	Year 1	Year 2	Year 3	Year 4	Year 5	Total
Relapse	€978,913.90	€204,018.06	€160,865.73	€86,459.86	€80,218.79	€1,510,476.33
Preterm birth	€300,991.04	€300,991.04	€300,991.04	€300,991.04	€300,991.04	€1,504,955.18
Vaccine	€-	€-	€-	€-	€-	€-
Total costs	€1,279,904.93	€505,009.09	€461,856.76	€387,450.89	€381,209.83	€3,015,431.51
**“With Vaccination” Scenario**						
	Year 1	Year 2	Year 3	Year 4	Year 5	Total
Relapse	€223,314.73	€46,541.62	€36,697.49	€19,723.66	€18,299.91	€344,577.41
Preterm birth	€68,663.58	€68,663.58	€68,663.58	€68,663.58	€68,663.58	€343,317.90
Vaccine	€2,171,939.82	€-	€-	€-	€-	€2,171,939.82
Total costs	€2,463,918.13	€115,205.20	€105,361.07	€88,387.24	€86,963.49	€2,859,83.13
**Differential Analysis**						
	Year 1	Year 2	Year 3	Year 4	Year 5	Total
Relapse	−€755,599.16	−€157,476.44	−€124,168.23	−€66,736.20	−€61,918.88	−€1,165,898.91
Preterm birth	−€232,327.46	−€232,327.46	−€232,327.46	−€232,327.46	−€232,327.46	−€1,161,637.28
Vaccine	€2,171,939.82	€-	€-	€-	€-	€2,171,939.82
Total costs	€1,184,013.20	−€389,803.89	−€356,495.69	−€299,063.66	−€294,246.33	−€155,596.38

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
