# Peer review of "HPV Vaccination in Women Treated for Cervical Intraepithelial Neoplasia: A Budget Impact Analysis"

_vaccines, 2021, doi:10.3390/vaccines9080816_

Round 1

Reviewer 1 Report

Please refer to the attached PDF

Author Response

We thank the Reviewer for his/her comments and suggestions. We have attached a file with the responses to all his/her comments.

Reviewer 2 Report

HPV, or human papillomavirus, is a common virus that can lead to 6 types of cancers later in life.  The progression of infection is related with the development of malignant lesions, particularly precancerous lesions of the uterine cervix (Cervical intraepithelial neoplasia - CIN). HPV vaccination is associated with a reduced risk of relapse in women surgically treated for HPV-related injuries. Therefore, in this study, Basile, Calabro and estimated the economic impact of extending HPV vaccination to this target population. This is an interesting and comprehensive study.

Comments:

  1. The authors should extend the conclusion.
  2. Moderate English language and style editing is required. Some sentences are not clearly written and should be revised

Author Response

(The authors gave the same response as above.)

Reviewer 3 Report

It seems informative and challenging study.

In the intro, many of parts are simple explanation of HPV infection and vaccines with general guideline of vaccination. 

I am not sure that the results from the cross sectional analysis could be  possible to be accepted without rigorous study design (cohort, or control group allocation).

Conclusions by the results seems to be not specific. 

Author Response

We thank the Reviewer for his/her comments and suggestions. We have attached a file with the responses to his/her comments.

Round 2

Reviewer 1 Report

The authors have successfully addressed all of my comments. They have significantly expanded the method section and added details and calculations which make the story more coherent. 

Reviewer 3 Report

I am not sure that the modifications could be acceptable.